# A Modified Proximal-Perturbed Lagrangian for Non-Convex Non-Smooth Representatives of Fairness Constraints

## Abstract

We study classification problems under fairness constraints and introduce an algorithmic framework designed to prevent discrimination against different groups. These problems are often reformulated as continuous constrained optimization problems and are typically solved using continuous relaxations (surrogates) of the fairness constraints. However, many current algorithms do not provide theoretical guarantees, which possibly is due to the resulting fairness constraints being both non-convex and non-smooth. We propose a novel primal-dual algorithm, based on a newly developed Lagrangian, that converges to a stationary solution of the reformulated problem. Our algorithm is not only efficient and robust, but it also enjoys strong performance guarantees on the fairness of its solutions. Furthermore, experimental results demonstrate that our algorithm is highly effective in terms of computational cost and fairness guarantees, outperforming related algorithms that use regularization (penalization) techniques and/or standard Lagrangian relaxation.

## 1 Introduction

Machine learning algorithms are increasingly used in enhancing human decision-making in sensitive domains. They can handle large amounts of data beyond human capacity with faster computation, increasing efficiency and accuracy. They also provide an alternative to human decision-making, which can be subjective and prone to biases, thus promising to enhance consistency in decision-making. Despite the efficiency and effectiveness of utilizing large datasets, they often lack the expected objectivity. Recent studies have revealed significant biases in these algorithmic decisions. For instance, Google's Ad-targeting algorithm showed a preference for recommending higher-paying executive positions to men more frequently than to women Datta et al. (2014). Similarly, an algorithm used in the U.S. criminal justice system incorrectly predicted African Americans to be twice as likely as white Americans to commit crimes again Chouldechova (2017).

Over the past decade, the development of fair classification algorithms has emerged as a critical topic in machine learning, due to the growing awareness of biases towards sensitive attributes in algorithmic decision-making. These algorithms are used in many applications, including the prediction of criminal recidivism (Dieterich et al., 2016; Flores et al., 2016), granting loans (Dedman et al., 1988), and job recommendation (Datta et al., 2014), to name a few. The objective of fair classification is to ensure that algorithms make decisions that are unbiased with respect to certain sensitive attributes in societal contexts including, but not limited to: gender, ethnicity, age (Angwin et al., 2022; Barocas & Selbst, 2016; Buolamwini & Gebru, 2018; Mehrabi et al., 2021). A variety of desired notions of fairness have been proposed; we refer to (Berk et al., 2021; Chouldechova & Roth, 2020; Mehrabi et al., 2021) for comprehensive discussions on fairness in machine learning and applications.

This work focuses on notions of fairness, which are widely used in classification applications. Popular group fairness notions include demographic parity (also known as statistical parity) (Dwork et al., 2012; Feldman et al., 2015), equal opportunity Hardt et al. (2016), and equalized odds (known as disparate treatment) (Hardt et al., 2016; Zafar et al., 2017). The underlying idea behind these notions is to balance the decisions of a classifier among the different sensitive groups. They can be incorporated into classification algorithms as constraints to mitigate biases.

## 1.1 RELATED WORK

The literature on algorithmic fairness is generally divided into three categories: pre-processing, in-processing, and post-processing (Mehrabi et al., 2021). Pre-processing methods focus on modifying the training data by removing correlations with sensitive features while preserving other data for training Zemel et al. (2013); Feldman et al. (2015); Samadi et al. (2018); Gordaliza et al. (2019). Post-processing methods adjust the model's predictions to meet fairness criteria, typically by modifying the decision boundary for specific subgroups Fish et al. (2016); Hardt et al. (2016) or using random classification for individuals from underprivileged groups Kamiran et al. (2012).

To control bias, fairness can be incorporated as constraints into optimization problems. A fair classification is thus formulated as a constrained optimization, aiming at minimizing the loss while ensuring that a fairness violation is kept within acceptable limits. This approach falls under the in-processing category. Recent works on fair classification have focused on developing algorithms to solve constrained optimization problems. Most fair classification algorithms use regularization techniques, where fairness constraints are penalized with certain regularization parameters (Agarwal et al., 2018; Berk et al., 2021; Celis et al., 2019; Donini et al., 2018; Menon & Williamson, 2018; Woodworth et al., 2017; Zafar et al., 2017). However, these regularization algorithms do not always provide provable fairness guarantees due to the non-convexity of the resulting optimization (e.g., statistical parity Dwork et al. (2012); Goel et al. (2018), and equalized odds Hardt et al. (2016); Menon & Williamson (2018)). Additionally, regularization algorithms can exhibit some disadvantages, such as: (i) they are often non-convex in nature or achieve convexity at the cost of probabilistic interpretation, and (ii) the performance of the algorithms is highly sensitive to the choices of hyper-parameters, leading to diverse results depending on different datasets Huang & Vishnoi (2019).

Another popular constrained optimization approach is to apply Lagrangian relaxation (Menon & Williamson, 2018; Cotter et al., 2019a; Narasimhan, 2018; Cotter et al., 2019b; Bendekgey & Sudderth, 2021; Cruz et al., 2022). Lagrangian methods allow for the incorporation of fairness constraints into the learning process by introducing multipliers that adjust the objective to account for fairness constraints. However, two main challenges arise when using Lagrangian with fairness constraints. The loss function is possibly non-convex, and the original fairness constraints are non-convex and non-differentiable. The non-differentiability can be effectively handled by replacing the original fairness constraints with smooth surrogates. The use of surrogates allows us to obtain solutions with optimality and provable fairness guarantees on the original constraints (Bendekgey & Sudderth, 2021; Yao et al., 2023). However, Lagrangian methods still have challenges posed by the non-convexity of fairness surrogates. The non-convexity of fairness constraints makes it challenging to ascertain whether a solution optimizes fairness and the failure of convergence.

## 1.2 OUR CONTRIBUTIONS

Motivated by the limitations in existing constrained optimization approaches to fair classification, we propose a novel algorithmic framework. This framework is based on a newly developed Lagrangian in Kim (2021; 2023), designed to tackle classification challenges with theoretical and performance guarantees. Our paper makes contributions to the literature on fair classification algorithms:

- Inspired by the Lagrangian formulation in Kim (2021) for equilibrium computation, we leverage artificial (perturbation) variables into the Lagrangian with dual smoothing, to derive strong concavity in the dual variables. This technique leads to an efficient primal-dual first-order algorithm for which we provide provable fairness guarantees. In particular, primal convergence naturally ensures feasibility (fairness) guarantees.

- Our algorithmic framework is flexible; it enables us to obtain fair classifiers under various fairness constraints, including non-convex non-smooth surrogates of the fairness constraints. It can also achieve approximate solutions that result in high accuracy of loss than prior work with high fairness.

- Our algorithm has a practical advantage with fixed parameters, except for the step size of the auxiliary multiplier. This feature simplifies implementation by removing extensive tuning of the parameters. It also consistently progresses towards balancing predictive accuracy and fairness guarantees. Experimental results show that our algorithm is efficient and performs favorably compared to related approaches that handle non-convex non-smooth constraints.

## 2 PRELIMINARIES

### 2.1 FAIR CLASSIFICATION

Let $\mathcal{D} = \{(x_i, s_i, y_i)\}_{i=1}^N$ be a set of $N$ training samples drawn independently from an unknown joint distribution of $(X, A, Y)$. $x_i \in X$ represents the predictive feature, $y_i \in \{0, 1\}$ is the target label, and $s_i \in \{a, b\}$ denotes the sensitive attribute. A parameterized classifier $f_\theta(x)$ predicts $y = 1$ if $f_\theta(x) > 0$. The goal of fair classification is to obtain a classifier $f_\theta(x)$ that is fair with respect to the given sensitive attribute while maintaining prediction accuracy. There are three fairness notions widely used for group fairness in classification: demographic parity Dwork et al. (2012); Agarwal et al. (2018), equal opportunity Hardt et al. (2016), and equalized odds (Hardt et al., 2016). Since demographic parity and equal opportunity are quite similar from a mathematical perspective (Lohaus et al., 2020), we focus on the two fairness notions of demographic parity and equalized odds:

**Demographic parity.** A classifier $f_\theta$ is fair for demographic parity if its predictions are independent of the sensitive attribute $A$: $\Pr(f_\theta(x) > 0 \mid s = a) = \Pr(f_\theta(x) > 0 \mid s = b)$.

**Equal opportunity.** $f_\theta$ is fair for equal opportunity if its predictions for positively labeled samples are independent of the sensitive attribute: $\Pr(f_\theta(x) > 0 \mid s = a, y = 1) = \Pr(f_\theta(x) > 0 \mid s = b, y = 1)$.

**Equalized odds.** $f_\theta$ holds equalized odds if its predictions are conditionally independent of the sensitive attribute: $\Pr(f_\theta(x) > 0 \mid s = a, y = j) = \Pr(f_\theta(x) > 0 \mid s = b, y = j), \forall j \in Y$.

In practice, we do not know the true distribution over $(X, A, Y)$ and only have access to training samples $\{(x_i, s_i, y_i)\}_{i=1}^N$. Furthermore, ensuring exact fairness may not be possible except in trivial cases (Woodworth et al., 2017; Friedler et al., 2021), or might come at a significant accuracy cost without guaranteeing fairness (Ji et al., 2020). Therefore, we consider the empirical fairness disparity $\Delta(\theta)$ based on the fairness score Wu et al. (2019), which is constrained to be within a specified $\varepsilon > 0$. The following definition provides the general form of this type of fairness disparity.

**Definition 1** (Empirical $\varepsilon$-fairness). A classifier $f_\theta$ satisfies a general group-based notion of empirical $\varepsilon$-fairness if $\Delta(\theta) := \left| \frac{1}{N_a} \sum_{i=1}^{N_a} \mathbb{I}\{f_\theta(x_i) > 0 \mid s_i, y_i\} - \frac{1}{N_b} \sum_{i=1}^{N_b} \mathbb{I}\{f_\theta(x_i) > 0 \mid s_i, y_i\} \right| \leq \varepsilon$, where $\mathbb{I}$ is the indicator function and $\varepsilon > 0$ is the *unfairness tolerance* parameter. A larger $\varepsilon$ permits greater fairness on a metric of interest, while a smaller $\varepsilon$ more tightly restricts the level of fairness.

The fairness-constrained empirical risk (loss) minimization can be formulated as (Donini et al., 2018; Goel et al., 2018):

$$\min_{\theta \in \Theta} \ F(\theta) := \frac{1}{N} \sum_{i=1}^N \ell_0(f_\theta(x_i), y_i) \quad \text{s.t.} \quad \Delta(\theta) \leq \varepsilon, \tag{1}$$

where $\ell_0 : \mathbb{R}^d \to \mathbb{R}$ is the loss, $F(\cdot)$ is the average predictive loss, and $\{(x_i, y_i)\}_{i=1}^N$ is a set of $N$ training samples. However, the constrained problem (1) is often intractable due to the non-convex and non-differentiable nature of $\mathbb{I}\{f_\theta(x) > 0 \mid \cdot\}$, making gradient-based algorithms inapplicable.

### 2.2 TRACTABLE OPTIMIZATION AND LAGRANGIAN RELAXATION

To address the intractability, these constraints can be replaced with suitable surrogates (Zafar et al., 2019; Cotter et al., 2019b; Lohaus et al., 2020; Bendekgey & Sudderth, 2021; Yao et al., 2023). We employ surrogates that are differentiable (or at least sub-differentiable) to enable the use of gradient-based algorithms. Specifically, let $\sigma : \mathbb{R} \to \mathbb{R}$ be a differentiable surrogate (continuous approximation) for the indicator function. For example, the indicator function $\mathbb{I}\{f_\theta(x) > 0 \mid s\}$ used for demographic parity can be replaced by $\sigma(f_\theta(x))$. We then set the tractable constraint:

$$\hat{\Delta}(\theta) := \left| \frac{1}{N_a} \sum_{s=a}^{N_a} \sigma(f_\theta(x)) - \frac{1}{N_b} \sum_{s=b}^{N_b} \sigma(f_\theta(x)) \right|, \tag{2}$$

Notice that the setting (2) is non-convex and non-smooth. Let $G(\theta) := \hat{\Delta}(\theta) - \varepsilon$ that is non-convex and non-smooth. In general, under the fairness constraints, finding a fair classifier for problem (1)

is approximately equivalent to solving the tractable continuous constrained optimization problem:

$$\min_{\theta \in \Theta} \ F(\theta) \quad \text{s.t.} \quad G(\theta) \leq 0, \tag{3}$$

where $F : \mathbb{R}^d \to \mathbb{R}$ is the differentiable loss; $G = (G_1, \ldots, G_m) : \mathbb{R}^d \to \mathbb{R}^m$ is a non-convex non-smooth mapping; and $\Theta \subseteq \mathbb{R}^d$ is a closed convex set. The corresponding Lagrangian is

$$\mathcal{L}(\theta, \lambda) = F(\theta) + \langle \lambda, G(\theta) \rangle,$$

where $\lambda \in \mathbb{R}^m_+$ is the Lagrange multipliers. Solving the constrained problem (3) via the Lagrangian $\mathcal{L}(\theta, \lambda)$ is equivalent with finding $(\theta^*, \lambda^*)$ that satisfies the KKT conditions as used in Hu & Chen (2020); Bendekgey & Sudderth (2021); Cruz et al. (2022):

**The KKT conditions**. A point $x^*$ is called a *KKT point* of problem (3) if there is $\lambda^*$ such that

$$\begin{cases} 0 \in \partial \mathcal{L}(\theta^*, \lambda^*) := \nabla F(\theta^*) + \partial G(\theta^*) \lambda^* + \mathcal{N}_\Theta(\theta^*) \\ \lambda^* \geq 0, \ \ G(\theta^*) \leq 0, \ \ \langle \lambda, G(\theta^*) \rangle = 0, \end{cases} \tag{4}$$

where $\mathcal{N}_\Theta(\theta^*) = \{ v \in \Theta \mid \langle v, \theta - \theta^* \rangle \leq 0, \ \forall \theta \in \Theta \}$ is the normal cone to $\Theta$ at $\theta^*$. Note that a suitable constraint qualification (CQ) is necessary for the existence of multipliers that satisfy the KKT conditions (e.g., MFCQ, CPLD, and others; see Bertsekas (1999); Andreani et al. (2022)).

## 3 PROXIMAL-PERTURBED LAGRANGIAN FRAMEWORK

In this section, we propose a new primal-dual framework that solves the problem of finding provably fair solutions. Given suitable surrogates of the fairness constraints, our method is guaranteed to find a classifier with a good level of fairness. To this end, we first introduce a novel Lagrangian that has a desirable structure for developing an efficient fair classification algorithm.

### 3.1 A VARIANT OF PROXIMAL-PERTURBED LAGRANGIAN

Motivated by the reformulation techniques in (Bertsekas, 1999; 2014), by employing *perturbation* variables $z \in \mathbb{R}^m$ with slack variables $u \in \mathbb{R}^m_+$, and letting $G(\theta) + u = z$ and $z = 0$, we first transform problem (3) into an extended equality-constrained formulation:

$$\min_{\theta \in \Theta, u \in \mathbb{R}^m_+, z \in \mathbb{R}^m} F(\theta) \quad \text{s.t.} \quad G(\theta) + u = z, \quad z = 0. \tag{5}$$

Clearly, for $z^* = 0$ and $u^* \geq 0$, the extended formulation (5) is equal to problem (3). For the equality constrained problem (5), we now define a variant of *the Proximal-Perturbed Lagrangian* (P-Lagrangian) introduced in Kim (2021) as follows:

$$\mathcal{L}_{\alpha\beta}(\theta, u, z, \lambda, \mu) := F(\theta) + \langle \lambda, G(\theta) + u - z \rangle + \langle \mu, z \rangle + \frac{\alpha}{2} \|z\|^2 - \frac{\beta}{2} \|\lambda - \mu\|^2, \tag{6}$$

where $\lambda \in \mathbb{R}^m$ is the multiplier (dual) associated with the constraint $G(\theta) + u - z = 0$ and $\mu \in \mathbb{R}^m$ is the *auxiliary multiplier* associated with the constraint $z = 0$, $\alpha > 0$ is a penalty parameter, and $\beta > 0$ is a proximal parameter.

In addition, observing that given $(\lambda, \mu)$, minimizing $\mathcal{L}_{\alpha\beta}$ with respect to $z$ gives:

$$z(\lambda, \mu) = (\lambda - \mu)/\alpha,$$

substituting $z(\lambda, \mu)$ into $\mathcal{L}_{\alpha\beta}(\theta, u, z, \lambda, \mu)$, yields the reduced P-Lagrangian:

$$\mathcal{L}_{\alpha\beta}(\theta, u, z(\lambda, \mu), \lambda, \mu) = F(\theta) + \langle \lambda, G(\theta) + u \rangle - \frac{1}{2\rho} \|\lambda - \mu\|^2, \tag{7}$$

where $\rho := \frac{\alpha}{1 + \alpha\beta}$. Note that $\mathcal{L}_{\alpha\beta}(\theta, u, z(\lambda, \mu), \lambda, \mu)$ is $\frac{1}{\rho}$-strongly concave in $\lambda$ (for fixed $\mu$) and hence there exists a unique maximizer, denoted by $\lambda(\theta, \mu)$. If we maximize $\mathcal{L}_{\alpha\beta}(\theta, u, z(\lambda, \mu), \lambda, \mu)$ with respect to $\lambda$, we obtain:

$$\lambda(\theta, \mu) = \operatorname*{argmax}_{\lambda \in \mathbb{R}^m} \mathcal{L}_{\alpha\beta}(\theta, u, z(\lambda, \mu), \lambda, \mu) = \mu + \rho(G(\theta) + u), \tag{8}$$

which is well-defined and will be used for the update of $\lambda_{k+1}$ in (12).

## 3.2 DESCRIPTION OF ALGORITHM

In this subsection, we present a gradient-based alternating algorithm that computes a stationary solution to problem (3). The steps of our proposed algorithm are described in Algorithm 1.

---

**Algorithm 1:** P-Lagrangian based Alternating Direction Algorithm (PLADA)

---

1: **Input:** fixed parameters $\alpha > 1$, $\beta \in (0, 1)$, $\rho = \frac{\alpha}{1+\alpha\beta}$, $0 < \eta < \frac{1}{L_F + 3\rho M_G^2}$ and $0 < \tau < \frac{1}{3\rho}$.

  initial $(\theta_0, u_0, z_0, \lambda_0, \mu_0)$, $\gamma_0 \in (0, 1]$.

2: **for** $k = 0, 1, \ldots, T$ **do**

3:   $\theta_{k+1} = \mathrm{argmin}_{\theta \in \Theta} \left\{ \langle \nabla F(\theta_k), \theta \rangle + \langle \lambda_k, G(\theta) \rangle + (1/2\eta)\|\theta - \theta_k\|^2 \right\}$

4:   $u_{k+1} = \Pi_U[u_k - \tau\lambda_k]$

5:   $\mu_{k+1} = \mu_k + \gamma_k \left( \frac{\lambda_k - \mu_k}{\rho} \right)$ with $\gamma_k = \min \left\{ \gamma_0, \frac{\rho\delta_k}{\|\lambda_k - \mu_k\|^2 + 1} \right\}$

6:   $\lambda_{k+1} = \mu_{k+1} + \rho(G(\theta_{k+1}) + u_{k+1})$

7:   $z_{k+1} = \frac{1}{\alpha}(\lambda_{k+1} - \mu_{k+1})$

8: **end for**

---

At each iteration, the algorithm updates $\theta$ by:

$$\theta_{k+1} = \mathrm{argmin}_{\theta \in \Theta} \left\{ \langle \nabla F(\theta_k), \theta \rangle + \langle \lambda_k, G(\theta) \rangle + 1/2\eta\|\theta - \theta_k\|^2 \right\}. \tag{9}$$

The update of $u$ is the projected gradient descent on $\mathcal{L}_{\alpha\beta}$ onto $U := [0, u_{\max}]$:

$$u_{k+1} = \mathrm{argmin}_{u \in U} \left\{ \langle \nabla_u \mathcal{L}_{\alpha\beta}(\theta_k, u_k, z_k, \lambda_k, \mu_k), u - u_k \rangle + 1/2\tau\|u - u_k\|^2 \right\} = \Pi_U[u_k - \tau\lambda_k], \tag{10}$$

where, without loss of generality, we can construct an upper $u_{\max} := B_G$ on $u_{k+1}$ as $\|G(\theta)\| \leq B_G$. The auxiliary multiplier $\mu$ is then updated by a gradient ascent scheme on $\mathcal{L}_{\alpha\beta}$:

$$\mu_{k+1} = \mu_k + \gamma_k(z_k + \beta(\lambda_k - \mu_k)) = \mu_k + \frac{\gamma_k}{\rho}(\lambda_k - \mu_k), \tag{11}$$

where we used the fact that $\nabla_\mu \mathcal{L}_{\alpha\beta}(\theta_{k+1}, z_k, \lambda_k, \mu_k) = z_k + \beta(\lambda_k - \mu_k)$ and $z_k = \frac{1}{\alpha}(\lambda_k - \mu_k)$; $\gamma_k > 0$ is the step-size defined by $\gamma_k = \min \left\{ \gamma_0, \rho\delta_k/(\|\lambda_k - \mu_k\|^2 + 1) \right\}$; and $\delta_k > 0$ in $\gamma_k$ is chosen to satisfy the following conditions: $\lim_{t \to \infty} \delta_k = 0$ and $\sum_{k=0}^{\infty} \delta_k = +\infty$. In the algorithm, we choose a decaying $\delta_k = \kappa \cdot (t+1)^{-1}$ with $\kappa > 0$, so that these conditions hold.

Next, the algorithm performs an exact maximization on the reduced P-Lagrangian (7) to update $\lambda$:

$$\lambda_{k+1} = \mu_{k+1} + \rho(G(\theta_{k+1}) + u_{k+1}). \tag{12}$$

The last step is to update $z$ via an exact minimization on $\mathcal{L}_{\alpha\beta}$ for given the updated $(\lambda_{k+1}, \mu_{k+1})$:

$$z_{k+1} = (\lambda_{k+1} - \mu_{k+1})/\alpha. \tag{13}$$

Note that a critical aspect of our algorithm is that the parameters $\alpha, \beta$, and the dual step size $\rho$ are constants and thus independent of the number of iterations $k$. In Appendix D.1, we demonstrate how robust the algorithm is with respect to the choices of $\alpha$ and $\beta$.

## 4 CONVERGENCE GUARANTEES

In this section, we present the convergence results of Algorithm 1. The structure of Algorithm 1 allows us to establish its convergence properties in a simple way. For the convergence analysis, we make the following standard assumptions:

**Assumption 1.** There exists a point $(\theta, \lambda) \in \Theta \times \mathbb{R}^m$ satisfying the KKT conditions (4).

**Assumption 2.** Given $\Theta \subseteq \mathbb{R}^d$, the gradient $\nabla F$ is $L_F$-Lipschitz continuous on $\Theta$. That is, there exist constants $L_F > 0$ such that $\|\nabla F(\theta) - \nabla F(\theta')\| \leq L_F \|\theta - \theta'\|$, $\forall \theta, \theta' \in \Theta$.

**Assumption 3.** $G$ is continuous with $\partial G(\theta) \neq \emptyset$ on $\Theta$, and there exists a constant $M_G > 0$ such that $\max_{\theta \in \Theta} \|\partial G(\theta)\| \leq M_G$.

**Assumption 4.** The domain $\Theta$ is compact, i.e., $D_\theta := \max_{\theta, \theta' \in \Theta} \|\theta - \theta'\| < \infty$.

**Assumption 5.** The iterates $\{\lambda_k\}$ are contained in a convex compact subset $\Lambda \subset \mathbb{R}^m$.

The assumptions above are standard in the optimization literature; see e.g., Boob et al. (2022); Huang & Lin (2023). Assumption 3 implies the Lipschitz continuity of $G$: $\|G(\theta) - G(\theta')\| \le M_G \|\theta - \theta'\|$, $\forall \theta, \theta' \in \Theta$. Problems with an unbounded $\Theta$ can be reformulated to satisfy Assumption 4. For example, if $F$ is bounded below and a coercive regularization $R$ is added, the problem with $F + R$ has a compact domain (see, e.g., Lu & Zhou (2023)). Moreover, Assumption 5 is commonly used in the convergence analysis of constrained optimization algorithms (Nocedal & Wright, 2006; Bertsekas, 2014; Birgin & Martínez, 2014; Hong et al., 2016; 2023; Na et al., 2023a;b).

## 4.1 MAIN RESULTS

In this subsection, we establish main convergence results for Algorithm 1. Building on several key properties of the proposed algorithm given in Appendix B (Lemmas 3 and 4), we show that the generated primal-dual iterates converge to a KKT point of problem (3).

**Theorem 1** (Primal convergence). *Suppose that Assumptions 1-4 hold. Let $\{(\theta_k, u_k, z_k, \lambda_k, \mu_k)\}$ generated by Algorithm 1, with the decaying sequence $\delta_k = \kappa \cdot (t+1)^{-1}$. Let $\{\mathbf{p}_k := (\theta_k, u_k, z_k)\}$ be the generated primal sequences. Then,*

$$\lim_{T \to \infty} \frac{1}{T} \sum_{k=0}^{T-1} \|\mathcal{G}_{\mathbf{p}}^{k+1}\|^2 = 0, \tag{14}$$

*where $\mathcal{G}_{\mathbf{p}}^{k+1} := (\mathcal{G}_\theta^{k+1}, \mathcal{G}_u^{k+1}, \mathcal{G}_z^{k+1}) \in \partial_{\mathbf{p}} \mathcal{L}_{\alpha\beta}(\theta_k, u_k, z_k, \lambda_k, \mu_k)$.*

Theorem 1, whose complete proof is provided in Appendix C.1, states that the $\tilde{\mathcal{O}}(1/T)$[1] rate of the primal convergence holds: the running-average stationarity (first-order optimality) residual is

$$\frac{1}{T} \sum_{k=0}^{T-1} \|\mathcal{G}_{\mathbf{p}}^{k+1}\|^2 = \mathcal{O}\left(\frac{\log(T)}{T}\right) = \tilde{\mathcal{O}}\left(\frac{1}{T}\right).$$

**Remark 1.** Invoking Lemma 4 and Theorem 1, we immediately obtain the following result:

$$\lim_{T \to \infty} \frac{1}{T} \sum_{k=0}^{T-1} \left(\|\theta_{k+1} - \theta_k\|^2 + \|u_{k+1} - u_k\|^2\right) = 0,$$

which implies the $\tilde{\mathcal{O}}(1/T)$ rate of the squared running-average successive difference of primal iterates:

$$\frac{1}{T} \sum_{k=0}^{T} \left(\|\theta_{k+1} - \theta_k\|^2 + \|u_{k+1} - u_k\|^2\right) = \tilde{\mathcal{O}}\left(\frac{1}{T}\right).$$

Note that Theorem 1 states the convergence in an ergodic sense, which involves averaging over the sequence of iterates or employing a randomized output selection from $T$ iterates. Thus, the primal iterates converge with $\tilde{\mathcal{O}}(1/T)$ in an ergodic sense.

We now show the feasibility guarantees for Algorithm 1. It suffices to prove that $\lim_{k \to \infty} \|\lambda_k - \mu_k\| = 0$. This result can be easily achieved by the auxiliary multiplier $\mu$ update.

**Theorem 2** (Fairness guarantees). *Under Assumptions $1-5$, let $\{(\theta_k, u_k, z_k, \lambda_k, \mu_k)\}$ be the sequence generated by Algorithm 1. Let the decaying sequence $\{\delta_k\}$ be chosen as in Theorem 1. Then, it holds that:*

$$\lim_{t \to \infty} \|\lambda_k - \mu_k\| = 0,$$

*and hence, we have that $G(\bar{\theta}) \le 0$, where $\bar{\theta}$ is a limit point of $\{\theta_k\}$.*

It is noteworthy that the above results suggest Algorithm 1 can reduce the fairness violation if controlling primal iterates $\{\theta_k\}$ and $\{u_k\}$ properly. By building on the convergent primal sequence and utilizing the definitions of $\lambda_{k+1}$ and $\mu_{k+1}$, we readily have the fairness guarantees. Equipped with Theorems 1 and 2, we immediately have the outer iteration complexity for Algorithm 1.

---

[1] The notation $\tilde{\mathcal{O}}(\cdot)$ suppresses all logarithmic factors from the big-$\mathcal{O}$ notation.

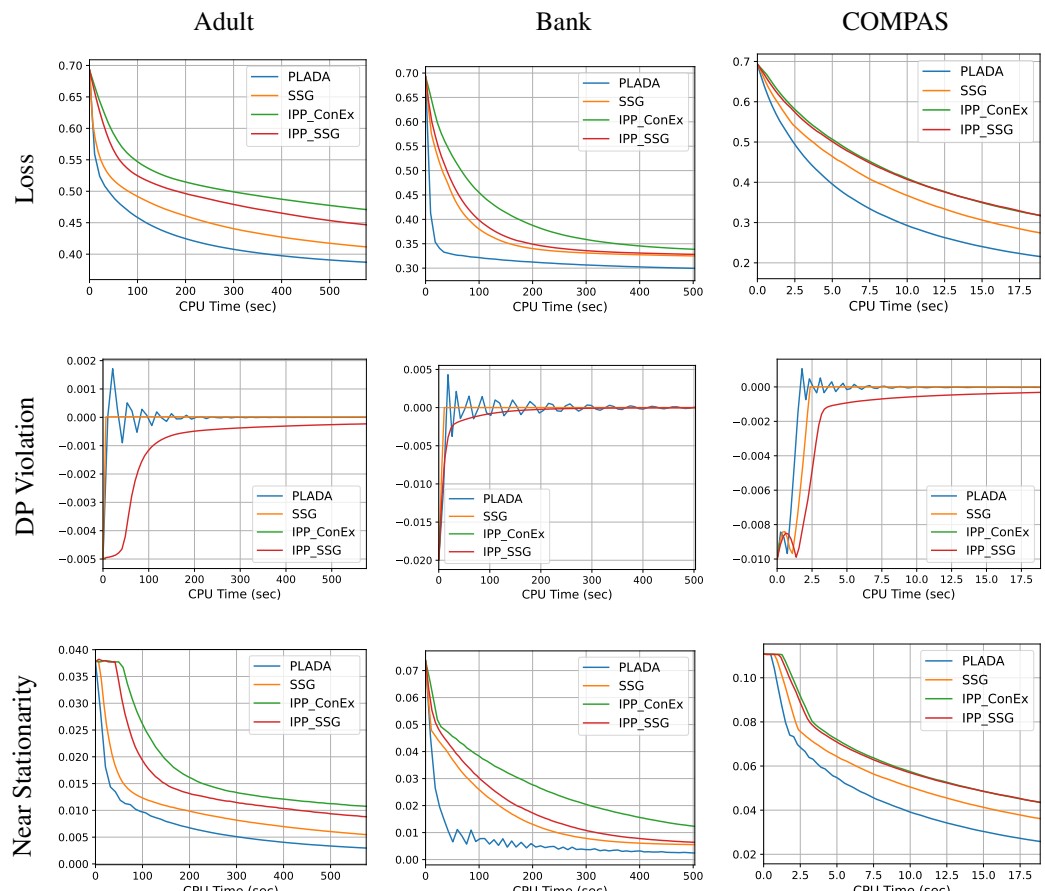

Figure 1: Comparison of the performance of PLADA, IPP-ConEx, IPP-SSG and SSG on the logistic loss (16) with demographic parity (DP) constraint (15). The results are presented in terms of their loss values, constraint violation and near stationarity (from top to bottom) on Adult, Bank and COMPAS datasets (from left to right) with respect to CPU time in seconds.

## 5 NUMERICAL EXPERIMENTS

We evaluate the empirical performance of our proposed algorithm on real-world datasets and compare with state-of-the-art algorithms that can handle non-convex non-smooth fairness constraints. Specifically, we evaluate the performance of PLADA against four benchmark algorithms: the single-loop switching subgradient (SSG) algorithm Huang & Lin (2023), two double-loop inexact proximal point (IPP) algorithms (IPP-ConEx Boob et al. (2022) and IPP-SSG Huang & Lin (2023)) and the multiplier model approach Narasimhan et al. (2020). For the benchmark algorithms, we followed the hyperparameter settings of Huang & Lin (2023) and Narasimhan et al. (2020), and we provide detailed descriptions of hyperparameters as well as additional experiments in the Appendix.

**Datasets.** We evaluate the performance of algorithms on real-world datasets in the field of algorithmic fairness: **Adult** (Kohavi et al., 1996), **Bank** (Moro et al., 2014), **COMPAS** (Angwin et al., 2022) and **Communities and Crime** (Redmond, 2009).

### 5.1 DEMOGRAPHIC PARITY CONSTRAINT

We start by considering the setting of non-convex non-smooth demographic parity constraint:

$$\widehat{\Delta}_D(\theta) = \left| \frac{1}{N_p} \sum_{i \in I_p} \sigma(\theta^\top x_i) - \frac{1}{N_u} \sum_{i \in I_u} \sigma(\theta^\top x_i) \right|, \tag{15}$$

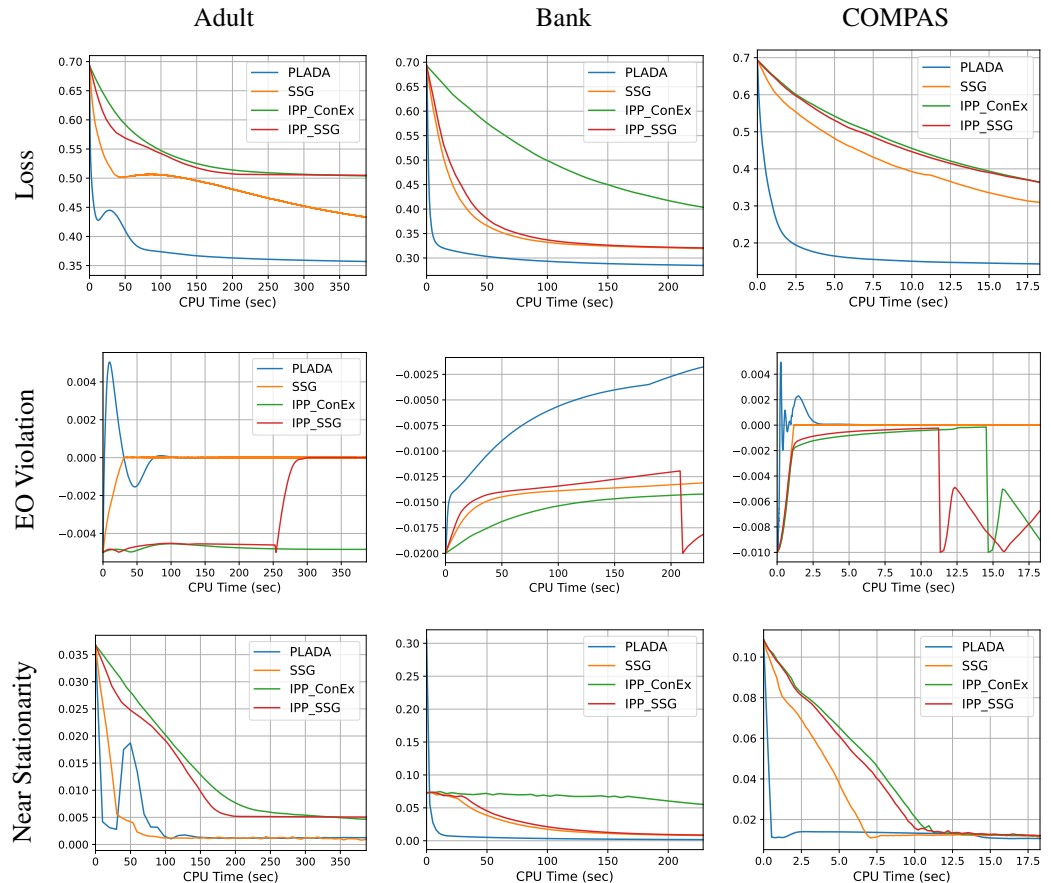

Figure 2: Comparison of the performance of PLADA, IPP-ConEx, IPP-SSG and SSG on the logistic loss objective (16) and the equalized odds (EO) constraint (17) with respect to CPU time.

where $I_p$ and $I_u$ denote the sets of protected and unprotected data indices, respectively, with corresponding sizes of $N_p = |I_p|$ and $N_u = |I_u|$. Equation (15) uses sigmoid $\sigma(\cdot)$ as a surrogate, making it weakly convex. And the objective is to optimize the logistic empirical loss:

$$F(\theta) = \frac{1}{N} \sum_{i=1}^{N} \log(1 + e^{-y_i \theta^\top x_i}), \tag{16}$$

Given that the logistic loss is smooth and convex, Figure 1 depicts the favorable behavior of each algorithm. Notably, our algorithm exhibits superior performance in the smooth and convex setting.

## 5.2 EQUALIZED ODDS CONSTRAINTS

While demographic parity (15) is a more widely accepted notion of fairness, equalized odds (17) is stricter and thus more challenging to optimize. Equalized odds aims to equalize the true positive rate and the false positive rate between protected and unprotected demographic groups.

$$\widehat{\Delta}_E(\theta) = \max\left( \left| \frac{1}{N_{pq}} \sum_{i \in I_{pq}} \sigma(\theta^\top x_i) - \frac{1}{N_{uq}} \sum_{i \in I_{uq}} \sigma(\theta^\top x_i) \right|, \right.$$

$$\left. \left| \frac{1}{N_{pu}} \sum_{i \in I_{pu}} \sigma(\theta^\top x_i) - \frac{1}{N_{uu}} \sum_{i \in I_{uu}} \sigma(\theta^\top x_i) \right| \right). \tag{17}$$

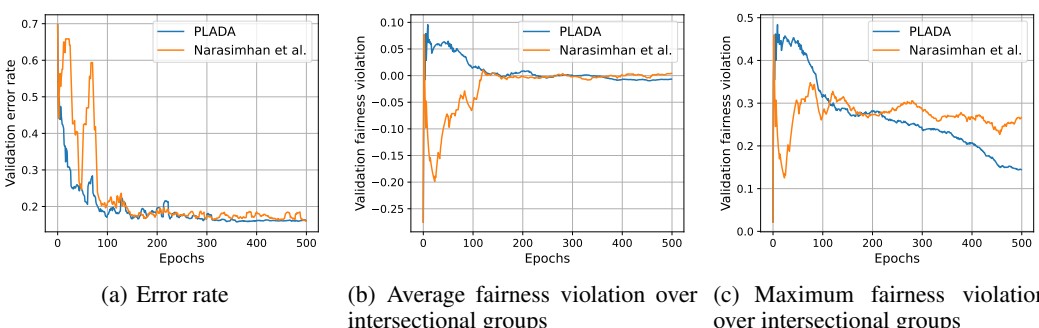

(a) Error rate

(b) Average fairness violation over intersectional groups

(c) Maximum fairness violation over intersectional groups

Figure 3: Comparison of the validation performance of PLADA and Narasimhan et al., Narasimhan et al. (2020) on the intersectional group fairness (18) versus Epochs.

While the compared algorithms support a single constraint, as described in (17), PLADA can handle multiple fairness constraints by alternatingly optimizing parameters such as $(u, z, \lambda, \mu)$ for each fairness constraint. The advantage of our algorithm over the other algorithms is most clearly illustrated in Figure 2.

### 5.3 EXTENSION TO INTERSECTIONAL GROUP FAIRNESS CONSTRAINTS ON NEURAL NETWORKS

Finally, we extend the experiment to an even more complex fairness problem by incorporating neural networks and intersectional group fairness. Specifically, we used a neural network with 5 hidden layers with ReLU activation for the classifier $f_\theta(\cdot)$. This makes both the objective and the constraints highly non-convex and non-smooth. Also, fairness over intersectional groups (18) is even stricter and more challenging than demographic parity (15) and equalized odds (17) in that the constraint spans over a large number of groups. In particular, we use the fairness constraint as the expectation over 535 intersectional fairness constraints:

$$\widehat{\Delta}_I(\theta) = \mathbb{E}_G \left[ \frac{1}{N_G} \sum_{i \in I_G} [1 - y_i f_\theta(x_i))]^+ - \frac{1}{N} \sum_{i=1}^{N} [1 - y_i f_\theta(x_i)]^+ \right], \tag{18}$$

where $G$ is a group uniformly sampled among all relevant groups and $[\cdot]^+$ represents a hinge function. For this problem, we used the Communities and Crime dataset Redmond (2009), which consists of 1,994 data points and 140 features, which aim to predict the per capita violent crimes of different communities in the US.

Notably, Figure 3 shows that PLADA outperforms the Lagrangian-based algorithm in Narasimhan et al. (2020), which uses a deep neural network with three hidden layers for updating the multipliers to ensure a bounded sequence. On the other hand, PLADA employs a simple updating scheme that guarantees the boundedness of the Lagrange multiplier sequence, leading to consistent fairness satisfaction.

## 6 CONCLUSIONS

We studied classification problems under fairness constraints, introducing an algorithmic framework to prevent discrimination across different groups. These problems are often reformulated as continuous constrained optimization tasks, using continuous relaxations of fairness constraints. Our novel primal-dual algorithm converges to a stationary solution at a rate of $\tilde{\mathcal{O}}(1/\sqrt{T})$, where $T$ represents the outer iterations. Experimental results demonstrated its effectiveness in terms of computational cost and fairness guarantees, outperforming related algorithms. Although our current analysis is limited to the use of deterministic/full (sub)gradient, extension to the stochastic setting is of interest. In Appendix D.3, we provide a preliminary application of our algorithm to stochastic gradients on a large dataset.

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

## A  NOTATION AND BASIC DEFINITIONS

Before proceeding with proofs of the lemmas and theorems, let us first provide the notation and basic definitions used in the proofs.

Let $\mathbb{R}^d$ denote $d$-dimensional Euclidean space with inner product $\langle \cdot, \cdot \rangle$ and the corresponding norm $\|\cdot\|$. The Euclidean norm of a matrix is also denoted by $\|\cdot\|$. We use $\mathbb{R}_+^m$ to denote the nonnegative orthant in $\mathbb{R}^m$ and denote the Jacobian matrix of $G$ at $\theta$ by $\partial G(\theta)$. The distance function between a vector $\theta$ and a set $\Theta \subseteq \mathbb{R}^d$ is defined by $\mathrm{dist}(\theta, \Theta) := \inf_{w \in \Theta} \|w - \theta\|$. For any set $\Theta \subseteq \mathbb{R}^d$, its indicator function $\mathbb{I}_\Theta$ is defined by $\mathbb{I}_\Theta = 0$ if $\theta \in \Theta$ and $+\infty$ otherwise. For any set $\Theta \subseteq \mathbb{R}^d$, its indicator function $\mathbb{I}_\Theta$ is defined by $\mathbb{I}_\Theta = 0$ if $\theta \in \Theta$ and $+\infty$, otherwise. Note that $\arg\min_{\theta \in \Theta} F(\theta) = \arg\min_{\theta \in \mathbb{R}^d} \{\varphi(\theta) := F(\theta) + \mathbb{I}_\Theta(\theta)\}$.

We recall some definitions about subdiffererential calculus (Rockafellar & Wets, 2009, Definition 8.3). Let $G_i : \mathbb{R}^d \to \mathbb{R} \cup \{+\infty\}$ be a proper and lower semicontinuous function. For each $\theta \in \Theta$, the *Frechet subdifferential* of $G$ of $\theta$ is given by

$$\widehat{\partial}G_i(\theta) := \left\{ d_k \in \mathbb{R}^d : \liminf_{w \to \mathbf{x}} \frac{G_i(w) - G_i(\theta) - \langle d, w - \theta \rangle}{\|w - \theta\|} \geq 0 \right\}.$$

The *limiting subdifferencial* (or simply the subdifferential) of $G_i$ at $\theta \in \mathbb{R}^d$ is defined as

$$\partial G_i(\theta) := \left\{ d \in \mathbb{R}^d : \exists \theta_k \to \theta \text{ and } d_k \in \widehat{\partial}G_i(\theta_k) \text{ with } d_k \to d \text{ as } k \to \infty \right\}.$$

The inclusion $\widehat{\partial}G_i(\theta) \subseteq \partial G_i(\theta)$ holds for each $\theta \in \Theta$ and we set $\widehat{\partial}G_i(\theta) = \partial G_i(\theta) = \emptyset$ for $\theta \notin \Theta$. Each $d \in \partial G_i(\theta)$ is called a subgradient of $G_i$ at $\theta$.

## B  PROOFS OF KEY PROPERTIES FORM MAIN RESULTS IN SECTION 4

Based on the structure of Algorithm 1, we first derive fundamental yet crucial relationships among the sequences $\{\lambda_k\}$, $\{\mu_k\}$, $\{\theta_k\}$, and $\{u_k\}$.

**Lemma 3.** *Let* $\{(\theta_k, u_k, z_k, \lambda_k, \mu_k)\}$ *be the sequence generated by Algorithm 1. Then,*

$$\|\mu_{k+1} - \mu_k\|^2 = (\gamma_k^2/\rho^2)\|\lambda_k - \mu_k\|^2 \leq \delta_k^2/2; \tag{19a}$$

$$\|\mu_{k+1} - \lambda_k\|^2 = (1 - (\gamma_k/\rho))^2 \|\mu_k - \lambda_k\|^2; \tag{19b}$$

$$\|\lambda_{k+1} - \lambda_k\|^2 \leq 3\rho^2 M_G^2\|\theta_{k+1} - \theta_k\|^2 + 3\rho^2\|u_{k+1} - u_k\|^2 + 3(\gamma_k^2/\rho^2)\|\lambda_k - \mu_k\|^2. \tag{19c}$$

*Proof.* From the $\mu$-update (11), we immediately obtain the relations in (19a):

$$\|\mu_{k+1} - \mu_k\|^2 = \frac{\gamma_k^2}{\rho^2}\|\lambda_k - \mu_k\|^2 \leq \frac{\delta_k^2}{\|\lambda_k - \mu_k\|^2 + 2 + (1/\|\lambda_k - \mu_k\|^2)} \leq \frac{\delta_k^2}{2}.$$

Subtracting $\mu_{k+1}$ from $\lambda_k$ yields

$$\|\lambda_k - \mu_{k+1}\| = \left\|\lambda_k - \mu_k - \frac{\gamma_k}{\rho}(\lambda_k - \mu_k)\right\| = \left(1 - \frac{\gamma_k}{\rho}\right)\|\lambda_k - \mu_k\|.$$

Squaring both sides of the above inequality yields the relation (19b).

By the $\lambda$-update (12), we have

$$\|\lambda_{k+1} - \lambda_k\| \leq \|\mu_{k+1} - \mu_k\| + \rho\|G(\theta_{k+1}) + u_{k+1} - G(\theta_k) - u_k\|$$
$$\leq \|\mu_{k+1} - \mu_k\| + \rho M_G\|\theta_{k+1} - \theta_k\| + \rho\|u_{k+1} - u_k\|,$$

which, along with $(a + b + c)^2 \leq 3(a^2 + b^2 + c^2)$ and (19a), provides the relation (19c). $\qquad\square$

**Lemma 4** (Approximate decrease of $\mathcal{L}_{\alpha\beta}$)**.** *Suppose that Assumptions $2-4$ are satisfied. Let* $\{\mathbf{w}_k := (\theta_k, u_k, z_k, \lambda_k, \mu_k)\}$ *be the sequence generated by Algorithm 1. Choose the step sizes $\eta$ and $\tau$ so that $0 < \eta < \frac{1}{L_F + 3\rho M_G^2}$ and $0 < \tau < \frac{1}{3\rho}$. Then, it holds that*

$$\mathcal{L}_{\alpha\beta}(\mathbf{w}_{k+1}) - \mathcal{L}_{\alpha\beta}(\mathbf{w}_k) \leq -C_1\|\theta_{k+1} - \theta_k\|^2 - C_2\|u_{k+1} - u_k\|^2 + \widehat{\delta}_k. \tag{20}$$

*where $C_1 := \frac{1}{2}\left(\frac{1}{\eta} - L_F - 3\rho M_G^2\right) > 0$, $C_2 := \frac{1}{2}\left(\frac{1}{\tau} - 3\rho\right) > 0$, and $\widehat{\delta}_k := \frac{\delta_k^2}{2\rho} + \frac{\delta_k}{\rho}$*

*Proof.* Notice first that

$$\mathcal{L}_{\alpha\beta}(\theta_k, u_k, z_k, \lambda_k, \mu_k) = F(\theta_k) + \langle \lambda_k, G(\theta_k) + u_k \rangle - \langle \lambda_k - \mu_k, z_k \rangle + \frac{\alpha}{2}\|z_k\|^2 - \frac{\beta}{2}\|\lambda_k - \mu_k\|^2$$

$$= F(\theta_k) + \langle \lambda_k, G(\theta_k) + u_k \rangle - \frac{1}{2\rho}\|\lambda_k - \mu_k\|^2$$

$$= \mathcal{L}_{\alpha\beta}(\theta_k, u_k, \widehat{z}(\lambda_k, \mu_k), \lambda_k, \mu_k),$$

where $\rho = \alpha/(1 + \alpha\beta)$, and thus

$$\mathcal{L}_{\alpha\beta}(\theta_{k+1}, u_{k+1}, z_k, \lambda_k, \mu_k) = \mathcal{L}_{\alpha\beta}(\theta_{k+1}, u_{k+1}, \widehat{z}(\lambda_k, \mu_k), \lambda_k, \mu_k).$$

Then the difference of two successive sequences of $\mathcal{L}_{\alpha\beta}$ can be divided into two parts:

$$\mathcal{L}_{\alpha\beta}(\theta_{k+1}, u_{k+1}, z_{k+1}, \lambda_{k+1}, \mu_{k+1}) - \mathcal{L}_{\alpha\beta}(\theta_k, u_k, z_k, \lambda_k, \mu_k)$$
$$= [\mathcal{L}_{\alpha\beta}(\theta_{k+1}, u_{k+1}, z_k, \lambda_k, \mu_k) - \mathcal{L}_{\alpha\beta}(\theta_k, u_k, z_k, \lambda_k, \mu_k)]$$
$$+ [\mathcal{L}_{\alpha\beta}(\theta_{k+1}, u_{k+1}, \widehat{z}(\lambda_{k+1}, \mu_{k+1}), \lambda_{k+1}, \mu_{k+1}) - \mathcal{L}_{\alpha\beta}(\theta_{k+1}, u_{k+1}, \widehat{z}(\lambda_k, \mu_k), \lambda_k, \mu_k)]. \tag{21}$$

Consider the first part (21). Since $\theta_{k+1}$ and $u_{k+1}$ are the solutions of subproblems (9) and (10), respectively, we have that for any $\theta \in \Theta$ and for any $u \in U$,

$$\langle \nabla F(\theta_k), \theta_{k+1} - \theta \rangle + \langle \lambda_k, G(\theta_{k+1}) - G(\theta) \rangle + \frac{1}{2\eta}\left(\|\theta_{k+1} - \theta_k\|^2 - \|\theta - \theta_k\|^2\right) \le 0, \tag{22}$$

and

$$\langle \nabla_u \mathcal{L}_{\alpha\beta}(\mathbf{w}_k), u_{k+1} - u \rangle + \frac{1}{2\tau}(\|u_{k+1} - u_k\|^2 - \|u - u_k\|^2) \le 0. \tag{23}$$

By taking $\theta = \theta_k$ in (22), $u = u_k$ in (23), and using $\nabla_u \mathcal{L}_{\alpha\beta}(\mathbf{w}_k) = \lambda_k$, we have

$$\langle \nabla F(\theta_k), \theta_{k+1} - \theta_k \rangle + \langle \lambda_k, G(\theta_{k+1}) - G(\theta_k) \rangle \le -\frac{1}{2\eta}\|\theta_{k+1} - \theta_k\|^2,$$

and

$$\langle \lambda_k, u_{k+1} - u_k \rangle \le -\frac{1}{2\tau}\|u_{k+1} - u_k\|^2.$$

By adding and subtracting the term $\langle \nabla F(\theta_k), \theta_{k+1} - \theta_k \rangle$, we obtain

$$\mathcal{L}_{\alpha\beta}(\theta_{k+1}, u_{k+1}, z_k, \lambda_k, \mu_k) - \mathcal{L}_{\alpha\beta}(\theta_k, u_k, z_k, \lambda_k, \mu_k)$$
$$= [F(\theta_{k+1}) + \langle \lambda_k, G(\theta_{k+1}) + u_{k+1} \rangle] - [F(\theta_k) + \langle \lambda_k, G(\theta_k) + u_k \rangle]$$
$$= \langle \lambda_k, G(\theta_{k+1}) - G(\theta_k) \rangle + \langle \lambda_k, u_{k+1} - u_k \rangle + [F(\theta_{k+1}) - F(\theta_k)]$$
$$= [\langle \nabla F(\theta_k), \theta_{k+1} - \theta_k \rangle + \langle \lambda_k, G(\theta_{k+1}) - G(\theta_k) \rangle]$$
$$\quad + [F(\theta_{k+1}) - F(\theta_k) - \langle \nabla F(\theta_k), \theta_{k+1} - \theta_k \rangle] + \langle \lambda_k, u_{k+1} - u_k \rangle$$
$$\le -\frac{1}{2}\left(\frac{1}{\eta} - L_F\right)\|\theta_{k+1} - \theta_k\|^2 - \frac{1}{2\tau}\|u_{k+1} - u_k\|^2. \tag{24}$$

Next, we derive an upper bound for the second part. We start by noting that

$$\mathcal{L}_{\alpha\beta}(\theta_{k+1}, u_{k+1}, \widehat{z}(\lambda_{k+1}, \mu_{k+1}), \lambda_{k+1}, \mu_{k+1}) - \mathcal{L}_{\alpha\beta}(\theta_{k+1}, u_{k+1}, \widehat{z}(\lambda_k, \mu_k), \lambda_k, \mu_k)$$
$$= \frac{1}{\rho}\langle \lambda_{k+1} - \lambda_k, G(\theta_{k+1}) + u_{k+1} \rangle - \frac{1}{2\rho}\left(\|\lambda_{k+1} - \mu_{k+1}\|^2 - \|\lambda_k - \mu_k\|^2\right).$$

Using the facts that $G(\theta_{k+1}) + u_{k+1} = \frac{1}{\rho}(\lambda_{k+1} - \mu_{k+1})$ and $\langle a, b \rangle = \frac{1}{2}\|a\|^2 + \frac{1}{2}\|b\|^2 - \frac{1}{2}\|a - b\|^2$ for any $a, b \in \mathbb{R}^m$, we have

$$\frac{1}{\rho}\langle \lambda_{k+1} - \lambda_k, \lambda_{k+1} - \mu_{k+1} \rangle = \frac{1}{2\rho}\left(\|\lambda_{k+1} - \lambda_k\|^2 + \|\lambda_{k+1} - \mu_{k+1}\|^2 - \|\mu_{k+1} - \lambda_k\|^2\right).$$

Hence,

$$\mathcal{L}_{\alpha\beta}(\theta_{k+1}, u_{k+1}, \widehat{z}(\lambda_{k+1}, \mu_{k+1}), \lambda_{k+1}, \mu_{k+1}) - \mathcal{L}_{\alpha\beta}(\theta_{k+1}, u_{k+1}, \widehat{z}(\lambda_k, \mu_k), \lambda_k, \mu_k)$$

$$\overset{(a)}{\le} \frac{1}{2\rho}\left(3\rho^2 M_G^2\|\theta_{k+1} - \theta_k\|^2 + 3\rho^2\|u_{k+1} - u_k\|^2 + 3\gamma_k^2\|\lambda_k - \mu_k\|^2\right) + \frac{1}{2\rho}\left(1 - (1 - \gamma_k)^2\right)\|\lambda_k - \mu_k\|^2$$

$$= \frac{1}{2}\left(3\rho M_G^2\|\theta_{k+1} - \theta_k\|^2 + 3\rho\|u_{k+1} - u_k\|^2\right) + \frac{3\gamma_k^2}{2\rho}\|\lambda_k - \mu_k\|^2 + \frac{1}{2\rho}\left(2\gamma_k - \gamma_k^2\right)\|\lambda_k - \mu_k\|^2$$

$$\overset{(b)}{\le} \frac{1}{2}\left(3\rho M_G^2\|\theta_{k+1} - \theta_k\|^2 + 3\rho\|u_{k+1} - u_k\|^2\right) + \frac{2\delta_k^2 + \delta_k}{\rho}, \tag{25}$$

where $(a)$ is from (19b) and (19c), and $(b)$ holds by $\gamma_k \|\lambda_k - \mu_k\|^2 \leq \frac{\delta_k}{1+(1/\|\lambda_k - \mu_k\|^2)} \leq \delta_k$. Combining (24) and (25) yields the desired result:

$$\mathcal{L}_{\alpha\beta}(\mathbf{w}_{k+1}) - \mathcal{L}_{\alpha\beta}(\mathbf{w}_k)$$
$$\leq -\frac{1}{2}\left(\frac{1}{\eta} - L_F - 3\rho M_G^2\right)\|\theta_{k+1} - \theta_k\|^2 - \frac{1}{2}\left(\frac{1}{\tau} - 3\rho\right)\|u_{k+1} - u_k\|^2 + \frac{2\delta_k^2 + \delta_k}{\rho},$$

which completes the proof. $\qquad\square$

## C  Proofs of Main Results in Section 4

Before presenting our main convergence results, we first derive an upper bound for the subgradient of $\mathcal{L}_{\alpha\beta}(\mathbf{w}_{k+1})$ in the primal variables. This subgradient, denoted by $\partial_{\mathbf{p}}\mathcal{L}_{\alpha\beta}(\mathbf{w}_{k+1})$, is expressed in terms of the iterates generated by Algorithm 1.

**Lemma 5** (Iterative error bound for subgradient of $\mathcal{L}_\rho$ in primal variables)**.** *Suppose that Assumptions 4 and 2 hold. Let the sequence $\{\mathbf{w}_k := (\theta_k, u_k, z_k, \lambda_k, \mu_k)\}$ be generated by Algorithm 1, and let $\{\mathbf{p}_k := (\theta_k, u_k, z_k)\}$ be the generated primal sequences. Then, there exists constant $d_1 > 0$ with $\mathcal{G}_{\mathbf{p}}^{k+1} := (\mathcal{G}_\theta^{k+1}, \mathcal{G}_u^{k+1}, 0) \in \partial_{\mathbf{p}}\mathcal{L}_{\alpha\beta}(\mathbf{w}_{k+1})$ such that*

$$\|\mathcal{G}_{\mathbf{p}}^{k+1}\| \leq D_{\mathbf{p}}\left(\|\theta_{k+1} - \theta_k\| + \|u_{k+1} - u_k\|\right) + (M_G + 1)\delta_k,$$

*where*

$$D_{\mathbf{p}} = \max\{L_F + 1/\eta + \rho(M_G^2 + M_G) + 1/\eta,\; \rho(M_G + 1) + 1/\tau\}.$$

*Proof.* Writing down the optimality condition for the update of $\theta_{k+1}$ in (9), we have

$$0 \in \nabla F(\theta_k) + \partial G(\theta_{k+1})^\top \lambda_k + \frac{1}{\eta}(\theta_{k+1} - \theta_k) + v,\;\; v \in \mathcal{N}_\Theta(\theta_{k+1}) \tag{26}$$

Using the subdifferential calculus rules, we have

$$\nabla F(\theta_{k+1}) + \partial G(\theta_{k+1})^\top \lambda_{k+1} + v \in \partial_\theta \mathcal{L}_{\alpha\beta}(\mathbf{w}_{k+1}) \tag{27}$$

By defining the quantity

$$\mathcal{G}_\theta^{k+1} = \nabla F(\theta_{k+1}) - \nabla F(\theta_k) + \partial G(\theta_{k+1})^\top(\lambda_{k+1} - \lambda_k) - \frac{1}{\eta}(\theta_{k+1} - \theta_k) \tag{28}$$

and using (26) and (27), we obtain that $\mathcal{G}_\theta^{k+1} \in \partial_\theta \mathcal{L}_{\alpha\beta}(\mathbf{w}_{k+1})$.

Next, define the quantity

$$\mathcal{G}_u^{k+1} := u_{k+1} - \Pi_U[u_{k+1} - \lambda_{k+1}],$$

which is equivalent to the *projected gradient* of $\mathcal{L}_{\alpha\beta}$ in $u$. It is a measure of optimality for the update of $u_{k+1}$ Nesterov (2012):

$$\widetilde{\nabla}_u \mathcal{L}_{\alpha\beta}(\mathbf{w}_{k+1}) := u_{k+1} - \operatorname*{argmin}_{v \in U}\left\{\langle \nabla_u \mathcal{L}_{\alpha\beta}(\mathbf{w}_{k+1}), v - u_{k+1}\rangle + \frac{1}{2}\|v - u_{k+1}\|^2\right\}$$
$$= u_{k+1} - \widetilde{u}_{k+1}.$$

where we define $\widetilde{u}_{k+1} := \operatorname{argmin}_{v \in U}\left\{\langle \nabla_u \mathcal{L}_{\alpha\beta}(\mathbf{w}_{k+1}), v - u_{k+1}\rangle + \frac{1}{2}\|v - u_{k+1}\|^2\right\}$.

From the update of $z_{k+1}$ in (13), we have

$$\nabla_z \mathcal{L}_{\alpha\beta}(\mathbf{w}_{k+1}) = -(\lambda_{k+1} - \mu_{k+1}) + \alpha z_{k+1} = 0.$$

Hence, we obtain

$$\mathcal{G}_{\mathbf{p}}^{k+1} := \begin{pmatrix} \mathcal{G}_\theta^{k+1} \\ \mathcal{G}_u^{k+1} \\ 0 \end{pmatrix} \quad \text{where} \quad \begin{pmatrix} \mathcal{G}_\theta^{k+1} & \in \partial_\theta \mathcal{L}_{\alpha\beta}(\theta_{k+1}, u_{k+1}, z_{k+1}, \lambda_{k+1}, \mu_{k+1}) \\ \mathcal{G}_u^{k+1} & = \widetilde{\nabla}_u \mathcal{L}_{\alpha\beta}(\theta_{k+1}, u_{k+1}, z_{k+1}, \lambda_{k+1}, \mu_{k+1}) \\ 0 & = \nabla_z \mathcal{L}_{\alpha\beta}(\theta_{k+1}, u_{k+1}, z_{k+1}, \lambda_{k+1}, \mu_{k+1}) \end{pmatrix}.$$

We derive an upper estimate for $\mathcal{G}_{\mathbf{p}}^{k+1}$. A direct calculation gives

$$
\begin{aligned}
\|\mathcal{G}_{\theta}^{k+1}\| &\leq \|\nabla F(\theta_{k+1}) - \nabla F(\theta_k)\| + (1/\eta)\|\theta_k - \theta_{k+1}\| + \|\partial G(\theta_{k+1})\|\|\lambda_{k+1} - \lambda_k\| \\
&\leq (L_F + 1/\eta)\|\theta_{k+1} - \theta_k\| + M_G\|\lambda_{k+1} - \lambda_k\| \\
&\leq (L_F + 1/\eta)\|\theta_{k+1} - \theta_k\| + \rho M_G^2\|\theta_{k+1} - \theta_k\| + \rho M_G\|u_{k+1} - u_k\| + M_G\delta_k \\
&\leq (L_F + 1/\eta + \rho M_G^2)\|\theta_{k+1} - \theta_k\| + \rho M_G\|u_{k+1} - u_k\| + M_G\delta_k
\end{aligned}
\tag{29}
$$

Next, we estimate an upper bound for the component $\mathcal{G}_u^{k+1}$. The first-order optimality condition implies that

$$
\langle \nabla_u \mathcal{L}_{\alpha\beta}(u_{k+1}) + (\widetilde{u}_{k+1} - u_{k+1}), u - \widetilde{u}_{k+1}\rangle \geq 0. \tag{30}
$$

Here, $\nabla_u \mathcal{L}_{\alpha\beta}(\mathbf{w}_{k+1})$ is denoted by $\nabla_u \mathcal{L}_{\alpha\beta}(u_{k+1})$. By the definition $u_{k+1}$ in (10), we have

$$
\left\langle \nabla_u \mathcal{L}_{\alpha\beta}(u_k) + \frac{1}{\tau}(u_{k+1} - u_k), u - u_{k+1}\right\rangle \geq 0, \tag{31}
$$

where $\nabla_u \mathcal{L}_{\alpha\beta}(u_k) = \nabla_u \mathcal{L}_{\alpha\beta}(\theta_k, u_k, z_k, \lambda_k, \mu_k)$ for simplicity. Combining (30) and (31), with settings $u = u_{k+1}$ in (30) and $u = \widetilde{u}_{k+1}$ in (31), yields

$$
\left\langle \nabla_u \mathcal{L}_{\alpha\beta}(u_k) - \nabla_u \mathcal{L}_{\alpha\beta}(u_{k+1}) + \frac{1}{\tau}(u_{k+1} - u_k) - (\widetilde{u}_{k+1} - u_{k+1}), \widetilde{u}_{k+1} - u_{k+1}\right\rangle \geq 0,
$$

equivalently,

$$
\left\langle \nabla_u \mathcal{L}_{\alpha\beta}(u_k) - \nabla_u \mathcal{L}_{\alpha\beta}(u_{k+1}) + \frac{1}{\tau}(u_{k+1} - u_k), \widetilde{u}_{k+1} - u_{k+1}\right\rangle \geq \|\widetilde{u}_{k+1} - u_{k+1}\|^2.
$$

By applying the Cauchy-Schwarz inequality and triangle inequality yields

$$
\left(\|\nabla_u \mathcal{L}_{\alpha\beta}(u_k) - \nabla_u \mathcal{L}_{\alpha\beta}(u_{k+1})\| + \frac{1}{\tau}\|u_{k+1} - u_k\|\right)\|\widetilde{u}_{k+1} - u_{k+1}\| \geq \|\widetilde{u}_{k+1} - u_{k+1}\|^2
$$

and

$$
\begin{aligned}
\|\nabla_u \mathcal{L}_{\alpha\beta}(u_k) - \nabla_u \mathcal{L}_{\alpha\beta}(u_{k+1})\| &\leq \|\lambda_k - \lambda_{k+1}\| \\
&\leq \rho M_G\|\theta_{k+1} - \theta_k\| + \rho\|u_{k+1} - u_k\| + \delta_k.
\end{aligned}
$$

Therefore,

$$
\|\mathcal{G}_u^{k+1}\| = \|\widetilde{u}_{k+1} - u_{k+1}\| \leq \rho M_G\|\theta_{k+1} - \theta_k\| + (\rho + 1/\tau)\|u_{k+1} - u_k\| + \delta_k. \tag{32}
$$

Combining (29) and (32), we obtain

$$
\|\mathcal{G}_{\mathbf{p}}^{k+1}\| \leq D_{\mathbf{p}}(\|\theta_{k+1} - \theta_k\| + \|u_{k+1} - u_k\|) + (M_G + 1)\delta_k,
$$

where $D_{\mathbf{p}} = \max\{L_F + 1/\eta + \rho(M_G^2 + M_G) + 1/\eta, \ \rho(M_G + 1) + 1/\tau\}$. This inequality, along with $\mathcal{G}_{\mathbf{p}}^{k+1} \in \partial \mathcal{L}_{\alpha\beta}(\mathbf{w}_{k+1})$, yields the desired result. $\qquad\square$

## C.1 PROOF OF THEOREM 1

*Proof.* From Lemma 4, we have

$$
C_{\mathbf{p}}\left(\|\theta_{k+1} - \theta_k\|^2 + \|u_{k+1} - u_k\|^2\right) \leq \mathcal{L}_{\alpha\beta}(\mathbf{w}_k) - \mathcal{L}_{\alpha\beta}(\mathbf{w}_{k+1}) + \widehat{\delta}_k, \tag{33}
$$

where $C_{\mathbf{p}} = \max\{C_1, C_2\}$. Using Lemma 5 and the fact $(a + b + c)^2 \leq 3(a^2 + b^2 + c^2)$, we have

$$
\|\mathcal{G}_{\mathbf{p}}^{k+1}\|^2 \leq 3D_{\mathbf{p}}^2(\|\theta_{k+1} - \theta_k\|^2 + \|u_{k+1} - u_k\|^2) + 3(M_G + 1)^2\delta_k^2,
$$

which, combined with (33), yields

$$
\|\mathcal{G}_{\mathbf{p}}^{k+1}\|^2 \leq \frac{3D_{\mathbf{p}}^2}{C_{\mathbf{p}}}\left(\mathcal{L}_{\alpha\beta}(\mathbf{w}_k) - \mathcal{L}_{\alpha\beta}(\mathbf{w}_{k+1}) + \widehat{\delta}_k\right) + 3(M_G + 1)^2\delta_k^2.
$$

Summing up the above inequalities over $k = 0, \ldots, T - 1$, we obtain

$$\sum_{k=0}^{T-1} \|\mathcal{G}_\mathbf{p}^{k+1}\|^2 \le \frac{3D_\mathbf{p}^2}{C_\mathbf{p}} \left( \mathcal{L}_{\alpha\beta}(\mathbf{w}_0) - \mathcal{L}_{\alpha\beta}(\mathbf{w}_T) + \sum_{k=0}^{T-1} \widehat{\delta}_k \right) + 3(M_G + 1)^2 \sum_{k=0}^{T-1} \delta_k^2$$

Since $\sum_{k=0}^\infty \delta_k^2 < +\infty$, we denote $B_\delta = \sum_{k=0}^\infty \delta_k^2$. Therefore,

$$\frac{1}{T} \sum_{k=0}^{T-1} \|\mathcal{G}_\mathbf{p}^{k+1}\|^2$$

$$\le \frac{\frac{3D_\mathbf{p}^2}{C_\mathbf{p}} (\mathcal{L}_{\alpha\beta}(\mathbf{w}_0) - \mathcal{L}_{\alpha\beta}(\mathbf{w}_T))}{T} + \frac{\frac{3D_\mathbf{p}^2}{C_\mathbf{p}} \sum_{k=0}^{T-1} \widehat{\delta}_k}{T} + \frac{3(M_G + 1)^2 \sum_{k=0}^{T-1} \delta_k^2}{T}$$

$$\le \frac{\frac{3D_\mathbf{p}^2}{C_\mathbf{p}} \left( \mathcal{L}_{\alpha\beta}(\mathbf{w}_0) - \underline{\mathcal{L}_{\alpha\beta}} \right)}{T} + \frac{\left( \frac{3D_\mathbf{p}^2}{2\rho C_\mathbf{p}} + 3(M_G + 1)^2 \right) \sum_{k=0}^{T-1} \delta_k^2}{T} + \frac{\frac{1}{\rho} \sum_{k=0}^{T-1} \delta_k}{T}, \quad (34)$$

where the second inequality holds by the the lower boundedness of $\mathcal{L}_{\alpha\beta}(\mathbf{w}_k)$, denoted by $\underline{\mathcal{L}_{\alpha\beta}}$, that is from the boundedness of generated sequences, and $\widehat{\delta}_k = \frac{\delta_k^2}{2\rho} + \frac{\delta_k}{\rho}$.

Note that given $\delta_k = \kappa \cdot (k + 1)^{-1}$ and $\kappa > 0$, for sufficiently large $T$, we know that

$$\sum_{k=0}^{T-1} \delta_k \approx \kappa^{-1} \log(\kappa T).$$

Since the last term on the right-hand side (RHS) of (34) dominates the other terms and $T$ grows faster than $\log(T)$, the RHS of (34) decreases to 0 as $T$ increase. Therefore, by taking the limit $T \to \infty$, we obtain

$$\lim_{T \to \infty} \frac{1}{T} \sum_{k=0}^{T-1} \|\mathcal{G}_\mathbf{p}^{k+1}\|^2 = 0 \text{ with the rate of } \mathcal{O}\left( \frac{\log(T)}{T} \right) = \tilde{\mathcal{O}}\left( \frac{1}{T} \right),$$

which proves that the ergodic primal convergence hold for Algorithm 1 in terms of the running-average stationarity residual. $\qquad \square$

## C.2 PROOF OF THEOREM 2

*Proof.* From the $\mu$-update (11), notice that $\mu_{k+1} = \mu_0 + \frac{1}{\rho} \sum_{t=0}^k \gamma_t(\lambda_t - \mu_t)$. Using the fact that $\|a\| - \|b\| \le \|a + b\|$ for any $a, b \in \mathbb{R}^m$, we have

$$\left\| \sum_{t=0}^\infty \gamma_t(\lambda_t - \mu_t) \right\| \le \|\mu_{k+1}\| + \|\mu_0\| < +\infty, \quad (35)$$

where the last inequality hold by the boundedness of $\{\mu_k\}$ from Assumption 5 together with the boundedness of sequence $\{(\lambda_k - \mu_k) := \rho(G(\theta_k) + u_k)\}$. The convergence of the sequences $\{\theta_k\}$ and $\{u_k\}$ to finite values $(\overline{\theta}, \overline{u})$, along with the definition of $\lambda_k = \mu_k + \rho(G(\theta_k) + u_k)$, implies that $\{\lambda_k - \mu_k\}$ is convergent to a finite value $(\overline{\lambda} - \overline{\mu})$.

We prove that $\{\lambda_k - \mu_k\} \to 0$ by contradiction. Assume that $\{\lambda_k - \mu_k\}$ does not converge 0, meaning there exists some $e \ne 0$ such that $\{\lambda_k - \mu_k\} \to e$ as $k \to \infty$. Since $\sum_{k=0}^\infty \gamma_k = \infty$, we see that

$$\left\| \sum_{k=0}^\infty \gamma_k(\lambda_k - \mu_k) \right\| = \infty,$$

which contradicts (35). This contradiction leads to the desired result that $\overline{\lambda} - \overline{\mu} = 0$. It directly follows the definitions of $\lambda_{k+1}$ and $u_{k+1}$ that

$$0 = \frac{1}{\rho} \left( \overline{\lambda} - \overline{\mu} \right) = G(\overline{\theta}) + \overline{u} \text{ and } \overline{u} \ge 0.$$

Hence, we have the feasibility of $\overline{\theta}$, namely, $G(\overline{\theta}) \le 0$. The above result, together with Theorem 1, implies that $\frac{1}{T} \sum_{k=0}^{T-1} \|\mathcal{G}_\mathbf{d}^{k+1}\|^2 = \tilde{\mathcal{O}}(1/T)$. $\qquad \square$

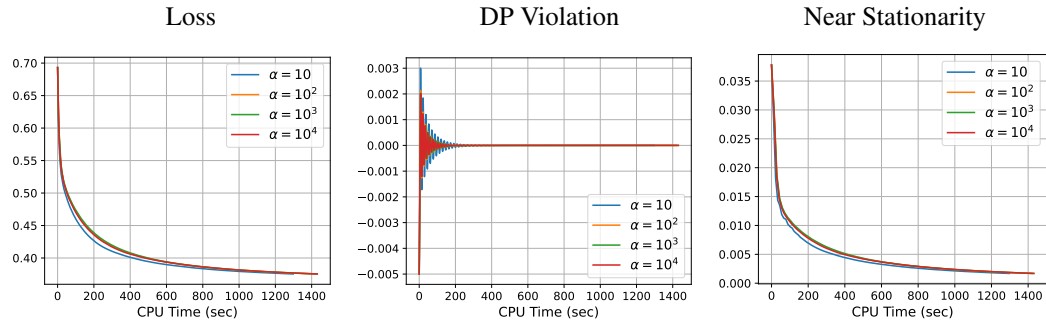

Figure 4: Comparison of the performance of PLADA with different $\alpha$ on the logistic loss objective with demographic parity (DP) constraint on Adult dataset. The results show the performance of PLADA is not sensitive to the value of $\alpha$ ($\beta = 0.1$ is fixed).

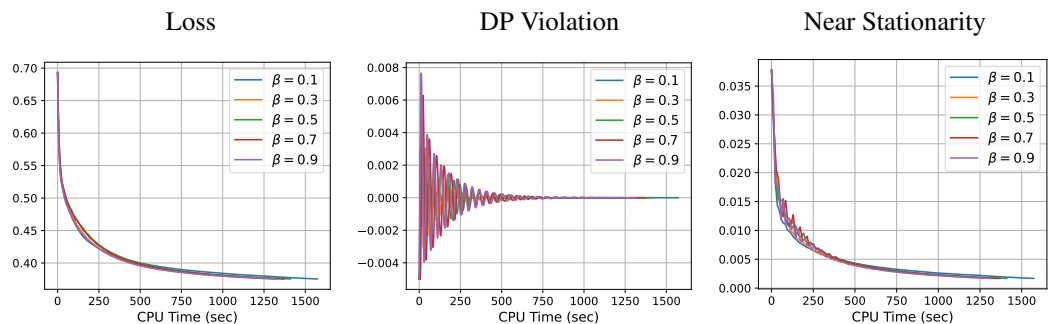

Figure 5: Comparison of the performance of PLADA with different $\beta$ on the logistic loss objective with demographic parity (DP) constraint on Adult dataset. The results show that the performance of PLADA is very slightly sensitive to the choice of $\beta$, as it affect dual parameter defined by $\rho = \frac{\alpha}{1+\alpha\beta}$ ($\alpha = 10$ is fixed).

# D  ADDITIONAL EXPERIMENTS

## D.1  HYPERPARAMETER SENSITIVITY

Although Algorithm 1 requires the selection of multiple hyperparameters $(\alpha, \beta, \rho, \eta, \tau)$, it is straightforward to select appropriate values for each hyperparameter. While $\eta$ is inevitably associated with any algorithm, $\rho$ and $\tau$ can be directly found by the values of $\alpha$ and $\beta$.

In this section, we provide empirical results on the sensitivity of our algorithm to the choices of the parameters $\alpha > 0$ and $\beta > 0$. First, Figure 4 demonstrates that the value of $\alpha$ does not have significant impact on the performance of the algorithm.

Additionally, Figure 5 shows that our algorithm demonstrates only a slight sensitivity to $\beta$, which is more evident in the near stationarity plot. Despite this slight sensitivity in $\beta$, our algorithm demonstrates sufficient robustness and can still provide solutions that minimize the objective and that remain feasible.

## D.2  CONVERGENCE OF DUAL VARIABLES

This section highlights empirical results that demonstrate the convergence of the dual variables, $\lambda$ and $\mu$, which further validate the results established by Theorem 2. The first row of Figure 6, demonstrates the converging behavior of $|\lambda - \mu|$ throughout multiple datasets. It can be easily seen from the figure, that the difference of $\lambda$ and $\mu$ converges to zero. The second and third rows depicts the individual convergence of $\lambda$ and $\mu$ respectively.

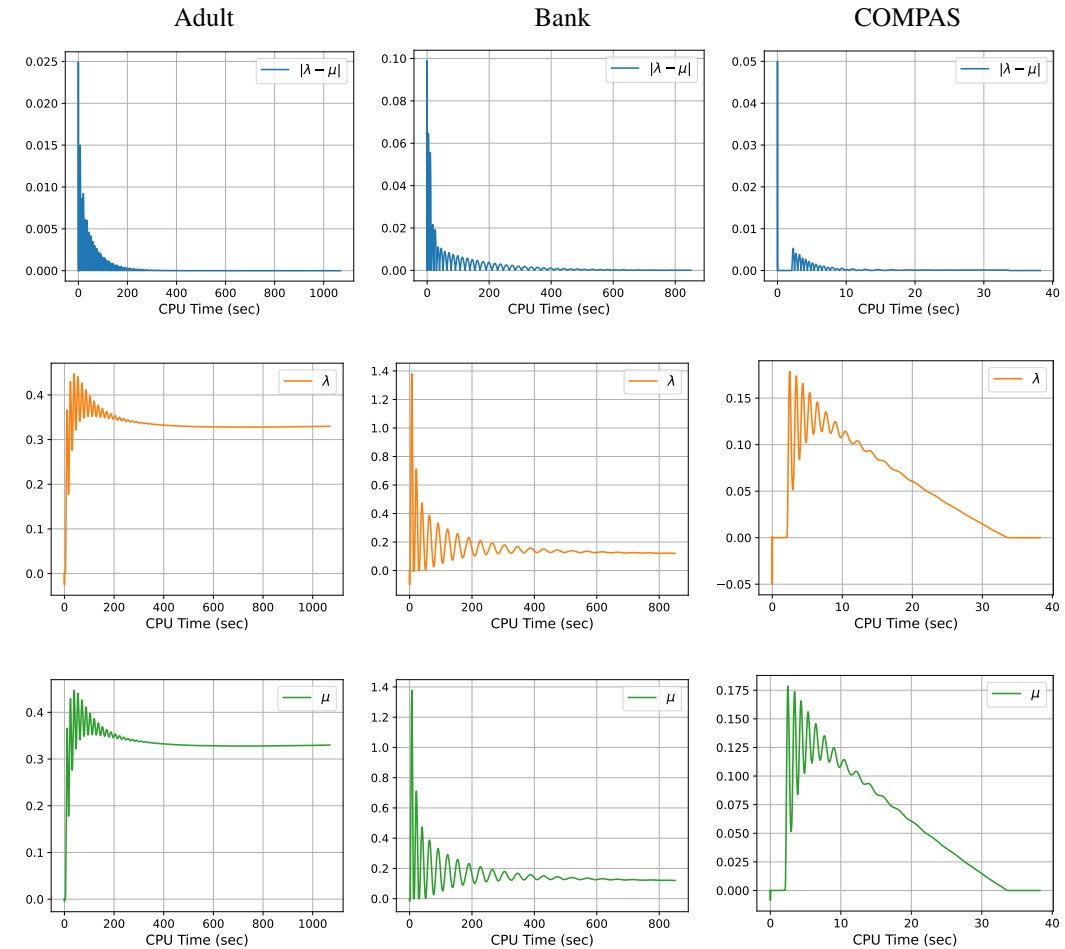

Figure 6: The values of $|\lambda - \mu|, \lambda, \mu$ of PLADA on the logistic loss objective with demographic parity (DP) constraint. The results show the converging behavior of the dual variables and their difference.

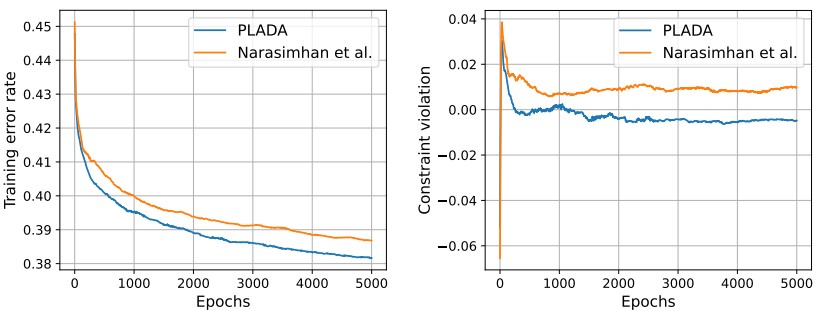

Figure 7: The average performance of PLADA and Narasimhan et al. (2020) on the ranking fairness versus Epochs after three repetitions. MSLR-WEB10K dataset has over 1.2M data points, from which over 470k pairs are created. PLADA achieves better constraint satisfaction with comparable error rate against approximate methods for the stochastic setting.

## D.3 HIGHLY STOCHASTIC SETTING

Another important setting to gauge the performance of our proposed algorithm is within the highly stochastic setting. In addition to dealing with stochastic mini-batches, we extended the experiment

to address ranking fairness. To do so, we leveraged the MSLR-WEB10K Dataset, which has over 470k pairwise constraints to satisfy.

We benchmarked our algorithm against Narasimhan et al. (2020). The results, as shown on Figure 7, demonstrate the effectiveness in finding more accurate classifier and the ability to better satisfy constraints, even under a highly stochastic setting.

# E  EXPERIMENT DETAILS

The description of the datasets used in the experiments are presented in Table 1.

| Dataset | n | d | Label | Sensitive Group |
|---|---|---|---|---|
| Adult (a9a) | 48,842 | 123 | Income | Gender |
| Bank | 41,188 | 54 | Subscription | Age |
| COMPAS | 6,172 | 16 | Recidivism | Race |
| Communities and Crime | 1,994 | 140 | Crime | Race |
| MSLR-WEB10K | 1.2 M | 136 | Relevance | Quality Score |

Table 1: Real-world fairness datasets used in experiments

Hyper-parameters of PLADA used in experiments are presented in Table 2. Note that we only used two hyper-parameter sets for 10 different problems, while our benchmark algorithms used different hyper-parameters for every datasets, objectives and constraints.

| Problem | $\eta_w$ | $\eta_u$ | $\alpha$ | $\beta$ | $\gamma_0$ |
|---|---|---|---|---|---|
| Models 5.1 and 5.2 | 0.001 | 0.1 | 10.0 | 0.1 | 0.1 |
| Neural network 5.3 | 0.1 | 0.01 | 10.0 | 0.5 | 0.1 |

Table 2: Hyper-parameters of PLADA used in experiments

Finally, the intersectional groups of Section 5.3 are created with ten thresholds on three criteria: the percentages of the Black, Hispanic and Asian populations. Among 1000 groups, 535 groups with memberships of more than 1% of data points form 535 constraints.

