# OpenReview forum: "A Modified Proximal-Perturbed Lagrangian for Non-Convex Non-Smooth Representatives of Fairness Constraints"
_ICLR.cc/2025/Conference — ICLR 2025 Conference Withdrawn Submission_

### Official Review · Reviewer_ykaY · 2024-10-30

**Soundness:** 1
**Presentation:** 2
**Contribution:** 3
**Rating:** 3
**Confidence:** 3

**Summary:**

The authors propose a new algorithmic approach for training fair classifiers, leveraging a general primal-dual optimization algorithm designed to approximate the saddle points of a Lagrangian function.

They demonstrate that common fair classification problems can be formulated such that an optimal fair classifier can be approximated using their optimisation algorithm.

Moreover, they provide theoretical guarantees for the proposed algorithms and present numerical experiments that illustrate the competitiveness of their approach compared to existing methods in the literature.

**Strengths:**

Up to my knowledge, the proposed approach is novel.
Moreover, it is backed by both theoretical and empirical evidence.

**Weaknesses:**

- The paper would benefit from additional explanations of the proposed approach in Section 3 (see Questions). The current description does not provide enough clarity to help the reader develop an intuitive understanding of the framework.

- I found the Main Results section difficult to follow, and I am not yet convinced of the soundness of the claims (see Questions below).


- The positioning of the paper is unclear: while the proposed approach has applications in fairness, it is not specific to fairness problems but rather a general optimisation framework. In this regard, I find the title, abstract, and introduction somewhat misleading.

**Questions:**

- The authors claim that the assumptions are standard in the optimization literature; however, after reviewing some of the cited papers, I could not find the exact same assumptions. Could the authors provide more precise references to the relevant literature? In particular for Assumption 5.

- Why is the problem in Eq. (3) tractable? As stated some lines above the constraint is non-convex and non-smooth. What do the authors mean by tractable?

- Could the authors explain the different terms in the Proximal-Perturbed Lagrangian in Eq. (6) ? In particular, why are the penalty and proximal terms added to the regular Lagrangian?

- Could the authors explain the different choices for the update rules (and the chosen order)? For instance, why is the parameter $\theta_k$ updated as in Eq. (9)?

- The authors state that they "show that the generated primal-dual iterates converge to a KKT point of problem (3)". The KKT conditions for problem (3) are given in (4) but I don't see the link with the obtained results in Theorem 1 and 2. Could the authors clarify this point?

- Unlike claimed by the authors, there is no rate of convergence in the statement of Theorem 1 (though one can be obtained from the proof). The authors should be more rigorous when describing (or stating) Theorem 1. Same comment applies to Remark 1.

- How does Theorem 2 guarantee the feasability guarantees for Algorithm 1?

- I cannot make the link between Lemma 4, Theorem 1 and Remark 1. Could the authors explain how they go form a bound on the norm of the sub-gradients to a bound on the iterates using Lemma 4?

- There is a running typo in the paper: the iteration number of the algorithm is sometimes denoted by $k$ and other times by $t$ (e.g., when the authors define $\delta_k = \kappa \cdot (t+1)^{-1}$.

**Details Of Ethics Concerns:**

The main results of the paper already appear in a different paper on arXiv that was pre-published some months ago.

---

### Official Review · Reviewer_A6Qa · 2024-10-31

**Soundness:** 1
**Presentation:** 2
**Contribution:** 1
**Rating:** 3
**Confidence:** 4

**Summary:**

This paper proposes a primal-dual algorithm for fair classification based on a modified proximal-perturbed Lagrangian. The main change from previous approaches is that they apply the algorithm from Kim (2021; 2023) to fair classification. They claimed higher computational efficiency. Experiments were provided on UCI datasets.

**Strengths:**

Pros:

The context and literature of fairness are well-referenced.

Experiments were provided including real data.

**Weaknesses:**

Cons:
1. The main difference between this work and previous work is that they reformulate the fair classification in (1) into their reformulated equation (3) by using a surrogate differentiable $\epsilon$-fairness instead of the indicator function in Definition 1. This step is fine, but the fairness classification is stemming from the equations in demographic parity/equalized odds and is replaced by the weaker surrogate in (2) because it can be solved by gradient-based method. Even if the proposed algorithm can solve their reformulation (3), there are no theoretical guarantees wrt to the original parity of the resulting solution.
2. Their main motivation is that this approach can directly handle the original non-smooth non-convex fairness-constrained problem. However, the reviewer is not convinced by this claim, as the paper also uses surrogate functions for the non-differentiable fairness constraints. If the proposed algorithm cannot solve the original non-convex problem exactly (due to the use of surrogates), it is unclear what theoretical or practical advantages it offers compared to the convex relaxations proposed in the literature, e.g., Donini et al., Celis et al., Goel et al.
3. The empirical evaluation is lacking. They benchmark with very few baselines compared to the rich literature on fair classification, notably, a solid comparison with the convex relaxation approaches is lacking, which is a central point given the paper's focus on non-convex optimization. More experimental details are needed for fair comparisons and reproducibility, e.g., the paper does not specify the hyperparameter tuning process for baselines, the values of ε. The absence of tables and standard deviation makes precise comparisons difficult. Furthermore, most of the empirical claims are only wrt CPU time but no thorough evaluation of convergence rates.

**Questions:**

1. The proposed algorithm can only handle two protected groups, what about multi-group problems?
2. In Eq. (2), the sum is over $s$ but no $s$ is found inside the sum.

---

### Official Review · Reviewer_mPD8 · 2024-11-02

**Soundness:** 2
**Presentation:** 3
**Contribution:** 1
**Rating:** 3
**Confidence:** 3

**Summary:**

This paper studies classification problems involving the formulation of fairness-constrained empirical risk (loss) minimization. The authors propose a modified proximal-perturbed Lagrangian based alternating direction algorithm (Algorithm 1) to the formulation, and present the convergence analysis. In the numerical experiments, when given a linear model, the authors conduct experiments that minimize the logistic empirical loss under demographic parity and equalized odds constraints. When given a neural network with RELU activations in the hidden layers, the authors conduct experiments that minimize the hinge empirical loss under the intersectional group fairness constraints.

**Strengths:**

The authors present plenty of fairness notions in Section 2 and correspondingly conduct multiple numerical experiments in Section 5.

**Weaknesses:**

The algorithmic framework proposed in Section 3 is weak. See Questions for the details.

**Questions:**

i) According to line 165-167 on page 4, $G=(G_1,\dots,G_m):\mathbb{R}^d\rightarrow\mathbb{R}^m$ is a non-convex non-smooth mapping. In Line 3 in Algorithm 3 from line 225-226 on page 5, the objective function of the subproblem on $\theta_{k+1}$ directly involves $\langle\lambda_k,G(\theta)\rangle$, which makes the subproblem itself a possible non-convex non-smooth problem. How to obtain $\theta_{k+1}$ in every iteration of Algorithm 1?

ii) $\theta_{k+1}$ in (9) on page 5 is used in (22) and (24) on page 15 in the convergence analysis. If we replace $\langle\lambda_k,G(\theta)\rangle$ by $\langle\nabla G(\theta_k)^\top\lambda_k,\theta\rangle$ in the subproblem, then (9) changes to
$$\theta_{k+1}=\text{arg}\min_{\theta\in\Theta}\Big(\langle\nabla_{\theta}\mathcal{L}_{\alpha\beta}(\theta_k,u_k,z_k,\lambda_k,\mu_k),\theta\rangle+\frac{1}{2\eta}||\theta-\theta_k||^2\Big)$$

$$\quad\quad=\Pi_\Theta\Big(\theta_k-\eta\nabla_{\theta}\mathcal{L}_{\alpha\beta}(\theta_k,u_k,z_k,\lambda_k,\mu_k)\Big).$$

We then need a trivial bound on the sequence of $\lambda_k$ to obtain the Lipschitz constant of $\nabla_{\theta}\mathcal{L}_{\alpha\beta}(\theta_k,\cdots)$ for all $k$ and adopt the constant step-size $\eta$ on $\theta$. The authors directly assume the boundedness of the sequence of $\lambda_k$ in Assumption 5 and say that this is standard in the optimization literature (line 271-274 on page 6). However, the boundedness of the dual sequence is proved as a result by assuming certain constraint qualifications, e.g., Theorem 5 (b) in Boob et al. (2023). How to justify Assumption 5?

---

### Official Review · Reviewer_3rEU · 2024-11-03

**Soundness:** 2
**Presentation:** 3
**Contribution:** 2
**Rating:** 5
**Confidence:** 2

**Summary:**

In this paper, a proximal-perturbed Lagrangian formulation was introduced to solve the problem of classification under relaxed fairness constraints. The method comes with a set of convergence and fairness guarantees provable under proper conditions. The numerical performance of the proposed algorithm has been evaluated on several non-convex non-smooth fairness constrained logistic regression tasks using benchmark datasets.

**Strengths:**

The idea of applying the proximal-perturbed Lagrangian formulation of Kim (2021) to fairness constrained classification is novel and interesting. The paper is generally well organized and clearly presented.

**Weaknesses:**

1. Concerning Theorem 1, the running-average stationarity residual is claimed to converge asymptotically. However, it was commented below the theorem that its proof actually suggests an $O(1/T)$ sublinear rate of convergence under the same conditions. I did not have a chance to check the proof in full details, but if this is really the case, then it is suggested to revise Theorem 1 to state the $O(1/T)$ convergence rate directly, as this would provide a stronger and more informative result. Also, this would help align the theorem statement with both the proof and the comments that follow.

2. In regard with Assumption 1 which requires the existence of primal-dual solutions satisfying KKT condition, it is not quite clear to me why such a condition should be reasonable for the non-convex and non-smooth Lagrangian formulation considered in this work. It is suggested to provide more detailed discussions on the validness of this assumption in the context.

**Questions:**

1. Can the result in Theorem 1 be presented in a non-asymptotic way as commented below the theorem (and as claimed in the concluding remarks as well)?

2. How to verify Assumption 1 (or strong duality) for the considered non-convex and non-smooth proximal-perturbed Lagrangian formulation?

---

### Official Review · Reviewer_3yQ8 · 2024-11-05

**Soundness:** 2
**Presentation:** 2
**Contribution:** 2
**Rating:** 5
**Confidence:** 5

**Summary:**

In this paper, the authors investigate classification problems subject to fairness constraints and propose an algorithmic framework. They specifically focus on continuous constraints with bounded subgradients. The authors begin by reformulating the tractable continuous constrained optimization problem using perturbation variables and slack variables, drawing inspiration from the work of Bertsekas. They then introduce a variant of the corresponding proximal-perturbed Lagrangian, substituting the perturbation variable
z with a function of the dual variables. In practice, this approach addresses a specific case of the original optimization problem.

**Strengths:**

They propose a proximal-perturbed Lagrangian frame to solve a constrained optimization and provide convergence guarantees.

**Weaknesses:**

1. $z(\lambda,\mu) =\frac{\lambda-\mu}{\alpha}$ is a special case.
2. In Equations (9) and (10), 2 second-order Taylor expansions are used to replaced the differential functions. Why you do not use the the differential functions themselves.
3. In this paper, the authors try to solve the tractable continuous constrained optimization problem (3). The result provided in Theorem 2 seems not so meaningful.

**Questions:**

See weakness.

---

### Note · Authors · 2024-11-25

I have read and agree with the venue's withdrawal policy on behalf of myself and my co-authors.